# How Work-Family Conflict Influenced the Safety Performance of Subway Employees during the Initial COVID-19 Pandemic: Testing a Chained Mediation Model

**DOI:** 10.3390/ijerph191711056

**Published:** 2022-09-03

**Authors:** Jingyu Zhang, Yao Fu, Zizheng Guo, Ranran Li, Qiaofeng Guo

**Affiliations:** 1CAS Key Laboratory of Behavioral Science, Institute of Psychology, Beijing 100101, China; 2Department of Psychology, University of Chinese Academy of Sciences, Beijing 100049, China; 3School of Transportation and Logistics, Southwest Jiaotong University, Chengdu 610031, China; 4National Engineering Laboratory of Integrated Transportation Big Data Application Technology, Chengdu 611756, China; 5National United Engineering Laboratory of Integrated and Intelligent Transportation, Southwest Jiaotong University, Chengdu 611756, China; 6Comprehensive Transportation Key Laboratory of Sichuan Province, Chengdu 610031, China; 7Chengdu Rail Transit Group Co., Ltd., Chengdu 610110, China

**Keywords:** safety performance, work-family conflict, job burnout, affective commitment

## Abstract

This study examined the impact of work-family conflict on subway employees’ safety performance during the initial wave of the COVID-19 pandemic. We proposed a chain mediation model in which job burnout and affective commitment play mediating roles in this process. Using questionnaire data from 632 Chinese subway employees during February 2020, structural equation modeling analyses were performed. The analyses showed that work-family conflict had a significant negative impact on subway employee safety performance. Moreover, job burnout completely mediated the influence of work-family conflict on safety performance, while affective commitment only partially mediated the influence of job burnout on safety performance. These findings suggest the important role played by Work-Family balance during the pandemic and contribute to a deeper understanding of the inner mechanisms. We also discussed several practical implications for organizations to reduce the negative impact of work-family conflict on safety performance.

## 1. Introduction

COVID-19 has greatly endangered public health since its commencement in early 2020 [1]. According to a recent estimation, over 514 million people have been infected, and over 6 million people have died of COVID-19 [2]. In addition to its threats to physical health, the pandemic also had great negative impacts on the mental health of the general population. Fear and anxiety toward the virus were prevalent [3,4]. Conflicting views toward the pathogenicity of the virus and the side effects of the vaccine have caused anger and even confrontation. Social distancing policies from city-level confinement to mask wearing may have caused difficulties in maintaining a healthy social support system. Given all these findings, it is not surprising to observe exponential growth in stress levels and symptoms of depression and anxiety [5].

While the general population may somehow take relief from reduced or distant work in this challenging period, the personnel in important public sectors and high-risk industries have to continue to work and guarantee safety. For example, subway personnel have to guarantee the safety of trains and passengers and prevent in-station infection when they have to deal with receiving less support from the outside and being exposed to a higher chance of infection. Such a situation may also influence other high-risk industry workers, including nuclear power plant operators and air traffic controllers. In this way, understanding how safety performance can be influenced by mental health-related variables during the pandemic is very important.

Safety performance involves rule-following behaviors consistent with certain safety regulations (safety compliance) and voluntary behaviors to promote safety beyond job responsibilities (safety participation) [6]. Past studies have found that safety performance is heavily influenced by stress-related variables such as work-family conflict and job burnout (for a comprehensive review, please see below for a full review). While these problems were also common during the pandemic [7,8,9,10], there are relatively few studies on how they were linked with safety performance during the outbreaks of COVID-19. In this study, we focus on several important variables that may have influenced the safety performance of subway workers during the first wave of the COVID-19 pandemic (from February 2020 to July 2020).

The first variable we took into consideration is work-family conflict. Work-family conflict is a form of inter-role conflict arising from the dual pressures of work and family, which are difficult to coordinate [11]. Because subway workers had to continue working during the pandemic, for preventive purposes, they had to work under additional restrictions on personal contact and quarantine, which could have undermined their fulfilment of family responsibilities. It has been shown that work-family conflict generally increased during the COVID-19 pandemic [12]. A survey revealed that 92% of frontline nurses showed “little concern” for their families amid the pandemic [13]. According to Work-Family boundary theory [14], there are relatively independent and limited resources in both the work and family fields. When either party, work or family, puts forward higher requirements, it may occupy the other party’s resources, break the original balance and cause conflict [15]. Past studies have found that work-family conflict can influence employees’ well-being and work outcomes [16]. Such a conflict may also undermine individuals’ ability to perform their duties safely and adequately. For example, researchers found that increased home–work conflict was associated with decreased compliance with safety rules among healthcare workers [17]. Similar results were also reported for truck drivers, train drivers and construction workers [18]. In an intervention study, Hammer (2016) found that training aimed to increase Work-Family balance skills and could increase contextual resources through supervisor support and more controllable work hours. All this led to improved personal resources, safety compliance and organizational citizenship behaviors in 30 health care facilities [19]. However, few studies have examined how work-family conflict influenced workers’ safety performance during the pandemic. Therefore, the first purpose of our study is to test whether work-family conflict decreases safety performance.

We also wanted to explore how this influence is mediated by other factors, such as job burnout. Job burnout is a state of physical fatigue and the exhaustion of mental resources caused by workers being unable to relieve the pressure caused by a high-intensity work environment for an extended period [20]. Many studies have found that job burnout became prevalent during the pandemic [21,22]. According to Work-Family boundary theory, work-family conflict depletes individual available resources, so people suffering from work-family conflict are more likely to experience job burnout. There is some evidence indicating that work-family conflict is an antecedent and a predictor of job burnout within various occupations [23,24], including law enforcement [25], nursing [26] and firefighting [27]. Some studies have also established a link between job burnout and safety performance. For example, Jia (2013) found that job burnout was negatively related to the safety performance of railroad locomotive drivers [28]. Smith (2018) found that firefighters’ job burnout can significantly impact their usage of protective equipment, their adherence to work norms and safety reporting and communication [29].

Based on these analyses, some researchers have suggested that job burnout can mediate the influence of work-family conflict on safety performance. They argue that burnout fully reflects the depletion of resources caused by work-family conflict [29,30]. Similar findings can also be found in miners [31,32], corporate workers [33], and healthcare workers [34]. However, some studies found evidence to support only a partial mediation effect. For example, Wu et al. (2018) found that, after considering the mediational role of job burnout, work-family conflict still had significant negative effects on the performance of construction workers [35]. One possible explanation is that work-family conflict may influence performance through other mechanisms that could be captured by job burnout. Therefore, the second purpose of the present study was to test the mediating role of job burnout and examine whether it plays a full meditation role or a partial mediation role.

Recent studies have also pointed out that the influence of job burnout on safety performance can be further mediated by more proximal motivational variables such as affective commitment. Affective commitment reflects the desire to work for an organization based on an employee’s inner recognition of the organization’s potential goals and values [36,37]. Some studies have found that job burnout reduces affective commitment. For employees in key positions, job burnout exposes them to stressors for an extended period, and they will not receive enough support to release their emotions [38]. The deprivation of valuable resources can lead to dissatisfaction, and, thus, individuals with job burnout are likely less satisfied with their work, creating cognitive dissonance and bringing about a situation in which they seek to reduce the negative effect on their work, eventually reducing their affective professional commitment. Job burnout has been found to reduce affective commitment for workers such as nurses [39], paramedics [40] and ambulance volunteers [41]. There is also evidence that affective commitment contributes to safety performance. For example, Smith (2019) found that affective organizational commitment was positively associated with both safety compliance and safety participation [42]. In a meta-analysis of 203 independent samples, Nahrgang et al. (2011) found that employee commitment is positively related to safety outcomes, including safety participation, safety communication and information sharing. They argue that employees’ organizational commitment contributes to increasing their willingness to comply with safety rules, so employees are willing to directly participate in organizational decision making, which ultimately improves safety performance [43]. Similarly, Curcuruto and Griffin (2018) have highlighted that affective commitment toward their organizations positively influences employees’ propensity to engage in safety citizenship behaviors, such as the support and protection of colleagues’ safety and health [44].

Based on these analyses, several studies have argued that affective commitment adequately mediates the effects of job burnout on safety performance. Sharma (2016) examined the factors that influence the affective commitment of nursing staff and its subsequent impact on their job performance. Based on the findings, healthcare institutions need to reduce the level of job burnout and create a supportive and fair working environment to enhance the level of affective commitment, thereby improving the performance of caregivers [45]. Some pieces of literature interpret affective commitment as a significant consequence of job burnout, which further affects job-related behaviors and employee activities [46,47]. Other studies have found evidence supporting only partial mediating effects. Prentice (2019) drew on the conservation of resources and the expectancy of motivation theories to study frontline employees working in the hospitality sector in the US. The results confirmed that job performance is related to job burnout, while employee commitment has a significant mediating effect on the relationship between burnout and performance [48]. Ozgur (2016) found that burnout significantly reduced the performance of healthcare workers and examined the negative correlation between the burnout subdimension and employee affective commitment [49]. Hence, the third purpose of the present study was to test the mediating role of affective commitment and examine whether it plays a fully mediating role or a partially mediating role.

In this study, we sought to examine the impact of work-family conflict on safety performance during the initial wave of COVID-19 for a group of subway workers. Based on previous studies, we constructed a chain mediating effect model to test the following hypotheses (see Figure 1).

**H1.** 
*Work-family conflict is significantly related to safety performance (i.e., safety participation and safety compliance);*


**H2.** 
*Job burnout mediates the association between work-family conflict and safety performance;*


**H3.** 
*Affective commitment mediates the association between job burnout and safety performance.*


## 2. Materials and Methods

### 2.1. Participants

We conducted an online survey to examine a total of 632 passenger attendants (28.96% men and 71.04% women). The mean age of these participants was 24.46 years (SD = 2.73), with an age range of 19–40 years. Their mean work experience was 2.54 years (SD = 2.58). Among them, 78.84% of the participants were unmarried, and 21.16% were married. In total, 75.52% of the participants had a college degree, 24.07% had a bachelor’s degree, and 0.41% had a graduate degree or higher. These subway employees need to work in shifts with a very irregular work schedule. Of these participants, those who slept less than 6 h occupied 27.53%, and those who slept 6–8 h occupied 68.82%. The detailed distributions regarding them are shown in Table 1.

### 2.2. Procedure

The data were collected from a Chinese metro company. With the help of metro company managers, we organized an online survey in July 2020. During this period, frontline employees resumed their normal work but were still at risk of coronavirus infection. All participants were required to complete an informed consent form before completing the online questionnaire to ensure that they were participating voluntarily.

### 2.3. Measures

#### 2.3.1. Work-Family Conflict

The work-family conflict was measured by an adapted version of the 10-item work-family conflict Questionnaire developed by Netemeyer [50]. It consists of two dimensions: Work-Family intervention (five items, e.g., “My work demands affect my home life”) and family–work intervention (five items, e.g., “I find it difficult to concentrate on my work due to the family stress”). The participants responded to each item on a seven-point Likert scale (1—completely disagree to 7—completely agree). We followed the approach suggested by Awwad (2022) by using the summed scores of all items to represent the construct of work-family conflict for further analyses [51]. The reliability (Cronbach’s α) of all 10 items was 0.91.

#### 2.3.2. Job Burnout

The Chinese version of the Maslach Burnout Inventory General Survey (MBI-GS) was used to measure job burnout [38,52]. The scale consists of fifteen items measuring emotional exhaustion, cynicism and reduced personal accomplishment. The participants were asked to rate the frequency of each stated feeling (e.g., “My work left me exhausted both physically and mentally”) on a seven-point scale from 1 (“never”) to 7 (“every day”). The higher the total score is, the greater the degree of job burnout. The Cronbach’s alpha was 0.80 for the overall scale. We followed Kang’s (2011) approach by using the summed scores of all items to represent the construct of job burnout for further analyses [53].

#### 2.3.3. Affective Commitment

Affective commitment was measured by the Chinese-translated version of the measurement tool proposed by Meyer and Allen (1991) [54]. This questionnaire consists of three items (e.g., “I am proud to be a member of my company”). The participants were asked to answer each question on a seven-point Likert scale from 1 (“strongly disagree”) to 7 (“strongly agree”). Higher total scores on this measure indicate a higher degree of affective commitment. In this study, Cronbach’s α was 0.87.

#### 2.3.4. Safety Performance

Safety performance was assessed using the scale developed by Griffin and Neal (2000) [6]. Three items were used to assess safety participation behaviors (e.g., “I am willing to make extra efforts to ensure workplace safety”), and three items were used to assess safety compliance behaviors (e.g., “I will use workplace protection and safety equipment in accordance with relevant regulations”). The participants rated their agreement with each statement along a seven-point Likert scale (“1 = strongly disagree” to “7 = strongly agree”). The higher the total score was, the better the safety performance was. The Cronbach’s alphas were 0.92 and 0.88 for safety participation and safety compliance, respectively.

#### 2.3.5. Control Variables

Gender and work experience were measured by demographic questions. To reduce common method bias, we used a strict criterion suggested by Podscooff (2003) by adding a control variable, self-esteem. Since this variable is both measured using the same measurement (self-report questionnaire) and theoretically linked with the other variables, using it as a control variable can produce a clean estimate of other interesting relationships. Self-esteem was measured using the Self-Esteem Scale developed by Rosenberg (1978) [55].

### 2.4. Data Analysis

Descriptive statistics were performed using SPSS 25 (Statistical Package for Social Sciences). Mediation analyses were conducted by a regression-based macro for SPSS 25.070 [56]. A bootstrapping procedure with 2000 replications was run to test the chain mediation model. Gender, work experience and self-esteem were used as control variables. work-family conflict, burnout, affective commitment and two safety performance measures were analyzed consecutively in a nested approach.

## 3. Results

### 3.1. Measurement Model

The measurement model was first examined using confirmatory factor analysis through AMOS 21.0. The eight-factor solution (two dimensions for work family conflict, three dimensions of job burnout and one dimension for affective commitment, safety participation and safety compliance, respectively) turned out to fit nicely with the data (χ^2^/df = 3.321, CFI = 0.949, TLI = 0.940, NFI = 0.929, RMSEA = 0.061).

For simplification, we used the total score of the work-family conflict by adding the 10 items of the two dimensions according to the method of Kuntsche (2021) [57]. Similarly, we used the total score of job burnout by adding all items of the three dimensions according to the method of Smith (2020) [58].

To examine the discriminant validity, we followed the method suggested by Fornell and Larcker (1981) [59]. All items correlated most strongly with their intended construct, and the square root of the AVE for the constructs was larger than any respective inter-construct correlations. These results provide evidence for discriminant validity (see Table 2). In summary, the measurement model demonstrated adequate reliability, convergent validity and discriminant validity.

### 3.2. Descriptive Statistics and Correlations among Variables

The results of descriptive statistics for each variable are shown in Table 3. The results show that metro employees basically have work-family conflict problems, with 34.49% and 10.13% of employees scoring between 26 and 35 and between 36 and 45 and 2.69% scoring above 45, which, together, represent almost half of the total participants. In terms of burnout, 37.34% and 3.8% of employees scored between 51 and 70 and between 71 and 90, with more than 40% of participants scoring in the medium-to-high range. Affective commitment, safety participation and safety compliance scores of more than 20 were found in 24.84%, 30.54% and 49.52% of employees, respectively.

The correlations provide provisional support for the hypotheses (see Table 2). Work-family conflict and job burnout are positively associated (r = 0.56, *p* < 0.01), while job burnout is negatively related to affective commitment (r = −0.58, *p* < 0.01), safety participation (r = −0.57, *p* < 0.01) and safety compliance (r = −0.40, *p* < 0.01). As expected, affective commitment is positively related to safety participation (r = 0.64, *p* < 0.01) and safety compliance (r = 0.41, *p* < 0.01).

### 3.3. Chain Mediation Effect Test

Table 4 presents the results of the mediation analyses. In predicting job burnout, work-family conflict was found to have a significant and positive association with job burnout (*p* < 0.001). In predicting affective commitment, job burnout was observed to show a significant and negative association with affective commitment (*p* < 0.001). In predicting safety participation, job burnout and affective commitment were significantly associated with safety participation (*p* < 0.001). In predicting safety compliance, job burnout and affective commitment were also significantly associated with safety compliance (*p* < 0.001).

Table 5 shows the chain mediating effect of job burnout and affective commitment on the relationship between work-family conflict and safety performance. For safety participation, the chain mediating effect of job burnout and affective commitment between work-family conflict and safety participation was significant (effect = −0.10, 95%CI = (−0.14, −0.08)). For safety compliance, the chain mediating effect of job burnout and affective commitment between work-family conflict and safety compliance was significant (effect = −0.06, 95%CI = (−0.09, −0.040).

Figure 2a shows the chain mediating effect of job burnout and the sequential chain mediating effect of job burnout and affective commitment on the association between work-family conflict and safety participation. All the paths in this model were significant (*p* < 0.001), except for the association between work-family conflict and affective commitment (B = −0.06, *p* > 0.05) and the association between work-family conflict and safety participation (B = 0.06, *p* > 0.05). Figure 2b shows the chain mediating effect of job burnout and the sequential chain mediating effect of job burnout and affective commitment on the association between work-family conflict and safety compliance. All the paths in this model were significant (*p* < 0.001), except for the association between work-family conflict and affective commitment (B = −0.06, *p* > 0.05) and the association between work-family conflict and safety compliance (B = −0.01, *p* > 0.05). The results showed that the chain mediating effect of job burnout and affective commitment was significant. In addition, job burnout fully mediated the relationship between work-family conflict and both types of safety performance. However, affective commitment only partially mediated the relationship between job burnout and safety performance.

## 4. Discussion

The main purpose of this study was to examine the impact of work-family conflict on safety performance among subway employees during the initial wave of the COVID-19 pandemic. We also sought to explore the mediating role of job burnout and affective commitment. Some findings are worth discussing.

First, we found that work-family conflict had a direct but negative influence on safety performance, which is in concordance with previous findings [17,18,29,60,61]. This result expands the literature in a new sample (subway employees) during a special period and is in concordance with many previous findings. The stress generated by work-family conflict may prevent employees from obtaining enough resources to achieve their work goals. Notably, we used a larger sample than previous studies, but the effect size in our study was still quite large. Cullen (2007) tested 243 healthcare workers and found that work-family conflict had a −0.23 effect on safety participation and a −0.27 effect on safety compliance [17]. Chu et al. (2020) studied 624 high-speed railway drivers and their 62 direct supervisors and found that the effect size of work-family conflict on safety participation and safety compliance was −0.15 and −0.16, respectively [18]. Wei (2016) explored the relationship between work-family conflict and safety participation in 494 high-speed railway drivers and found that work-family conflict was a strong predictor of safety engagement, with a predicted value of −0.16 [60]. This study found that work-family conflict still had a high effect size on safety participation and safety compliance (effect = −0.30, −0.24) in a survey of 632 subway frontline employees. The mean value of work-family conflict in the previous study was 3.30 [18] and 3.03 [60]. The average work-family conflict value of this study was 31.91, and the average value of the 10 items was consistent with the work-family conflict average value of the previous study. The outbreak of the COVID-19 pandemic may have aggravated the negative impact of work-family conflict, as it can reduce the chance for constructive communication, which is believed to decrease the negative consequences of work-family conflict.

Second, this study found that job burnout fully mediated the relationship between work-family conflict and safety performance. As job burnout reflects the magnitude of resource depletion, this finding helps explain the inner mechanisms by which work-family conflict influences safety performance. work-family conflict can further reduce the limited resources of employees, which will in turn undermine their safety behaviors. However, unlike many previous studies, job burnout in this study completely mediated the relationship between work-family conflict and safety performance. One possible explanation for this is that, due to the impact of the pandemic, job burnout has become so severe and widespread that it overshadows other mechanisms that affect safety performance. During the pandemic, the severe situation of pandemic prevention and unrelenting characteristics increased the psychological pressure on employees, and it is easy for employees to become tired. Job burnout in this environment may cause employees to ignore some safety and pandemic prevention norms, thus showing behaviors that are not conducive to enterprise safety [62]. Therefore, in short, this study emphasizes that job burnout can adequately mediate the effect of work-family conflict on safety performance.

Finally, we found that affective commitment partially mediates the influence of job burnout on safety performance. Defined as an individual’s emotional attachment to an organization, affective commitment is linked to the sense of belonging and motivations to work for the organization, such as low turnover intentions, and a greater willingness to perform organizational citizenship behaviors [63,64]. However, employees may disengage and reduce their affective commitment to their organizations to prevent the further loss of resources when they feel greatly depleted (burnout). This can be considered a withdrawal response in coping with job burnout. Given that affective commitment only partially explains the burnout–performance link, some other potential mechanisms might be worth further exploring. For example, it might be related to reduced cognitive capacities. When subway employees feel burnout, they could be more likely to experience problems such as reduced concentration, diminished self-control and memory loss. They are less likely to follow standard operating procedures or safely perform standard work practices. Future studies may benefit from investigating this and other possible mechanisms.

### 4.1. Implications

Some implications can be derived from these findings. First, they highlight the importance of work-family conflict during the pandemic. To promote safety during the pandemic, organizations should pay more attention to employee work-family conflict and develop specific plans and measures to reduce it. First, organizations should pay more attention to the needs and psychological states of their employees. In this way, they can find potential risk factors in advance. Second, when these risks are identified, organizations should provide corresponding measures to reduce the work-family conflict experienced by their employees. For example, they can offer possible training for workers to better deal with such problems. They can also enact certain policies to promote Work-Family balance, such as providing daycare centers for workers who are parents of small children.

Second, it emphasizes that burnout plays an important role in the impact of work-family conflict on safety performance. Organizations should pay more attention to employee burnout and actively seek effective measures to improve employees’ work status. First, organizations should pay attention to employees’ needs, value their pursuit of values and provide necessary work information and resources in a timely manner to strengthen employees’ sense of organizational support. Secondly, the organization can make corresponding policies to improve the burnout situation of employees, such as the rational optimization of the shift work system and letting employees have a good sleep so that they can work with full energy.

### 4.2. Limitations

We have to mention several limitations before making further conclusions. First, the cross-sectional nature of this study prevents any causal conclusions from being drawn. While a chained mediation was found, further experimental and longitudinal studies are required to examine its validity. Second, although we used certain statistical methods (e.g., using covariates) to control for common method bias, future studies may benefit from using data from different sources (e.g., supervisor-rated performance and partner-rated work-family conflict). Finally, although we expanded the stream of research by investigating a previously unstudied occupation (subway personnel) during a particular period (the initial wave of COVID-19), whether it can be generalized to other job categories and another time remains to be determined.

## 5. Conclusions

During the first few months of COVID-19, subway employees’ safety performance was negatively affected by work-family conflict. This study sheds more light on the inner mechanisms by which work-family conflict influences safety performance. The model results showed a direct but negative effect of work-family conflict on safety performance, with job burnout fully mediating the effect of work-family conflict on safety performance and affective commitment partially mediating the effect of job burnout on safety performance. It also calls for practitioners to understand the negative consequences of work-family conflict and seek out methods to reduce it.

## Figures and Tables

**Figure 1 ijerph-19-11056-f001:**
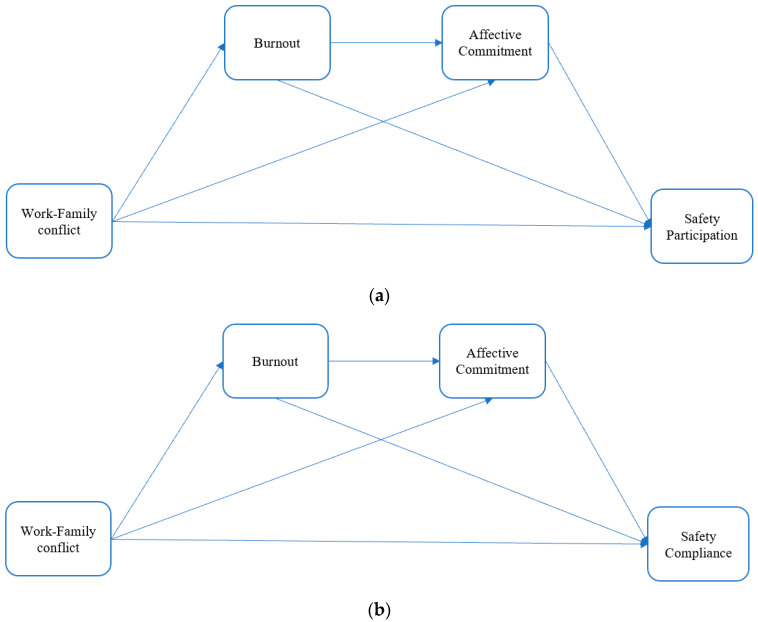
The proposed chain mediation model of the impact of work-family conflict on safety performance. (**a**) Safety performance: Safety participation. (**b**) Safety performance: Safety compliance.

**Figure 2 ijerph-19-11056-f002:**
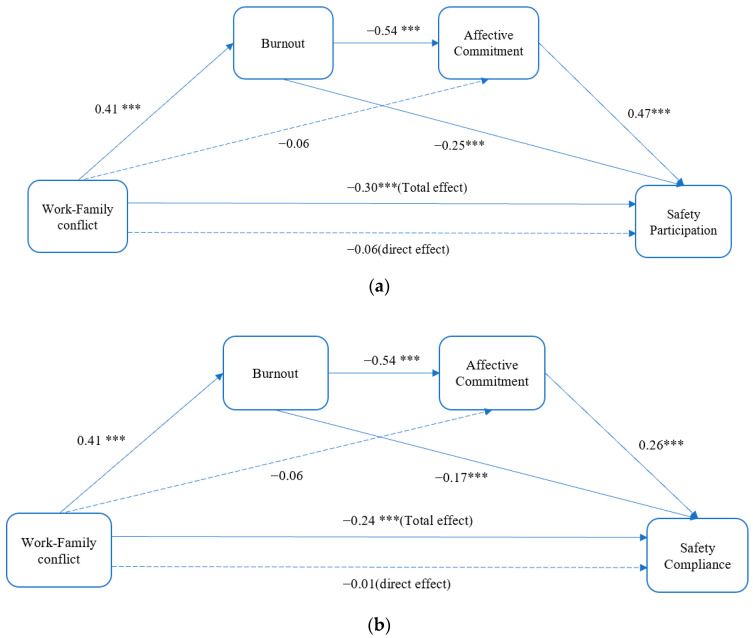
Tests of the chain mediation model. (**a**) Safety performance: Safety participation. *** Significant at *p* < 0.001. (**b**) Safety performance: Safety compliance. *** Significant at *p* < 0.001.

**Table 1 ijerph-19-11056-t001:** Demographic characteristics of the respondents.

Characteristic	Variable	%
Gender	Female	71
	Male	29
Age	19–25	73.89
	26–30	21.99
	31–46	4.12
Work experience	0–3 years	74.52
	4–6 years	15.35
	>6 years	10.13
Marital status	Unmarried	78.84
	Married	21.16
Level of education	College	75.52
	Undergraduate	24.07
	Graduate and above	0.41
Sleep hours	<6 h	27.53
	6–8 h	68.82
	9–10 h	3.16
	>10 h	0.49

**Table 2 ijerph-19-11056-t002:** Means, standard deviations and zero-order correlations among all variables.

Constructs	Mean	SD	1	2	3	4	5	6	7	8
1. Gender	0.29	0.45	--							
2. Work experience	2.54	2.59	0.03	--						
3. Self-esteem	30.61	4.58	0.12 **	0.03	--					
4. work-family conflict	31.91	12.28	0.10 *	0.07	−0.35 **	0.76				
5. Job burnout	30.68	14.22	−0.11 **	0.09*	−0.58 **	0.56 **	0.69			
6. Affective commitment	17.37	3.16	0.01	−0.13**	0.34 **	−0.36 **	−0.58 **	0.89		
7. Safety participation	17.89	2.84	0.11 **	−0.03	0.43 **	−0.30 **	−0.57 **	0.64 **	0.86	
8. Safety compliance	19.46	1.89	0.04	0.03	0.33 **	−0.24 **	−0.40 **	0.41 **	0.55 **	0.91

Notes: √ AVE estimates for latent variables are presented on the diagonal. * *p* < 0.05; ** *p* < 0.01.

**Table 3 ijerph-19-11056-t003:** Descriptive statistics of the variables.

Variables	Range	Categories	N	%
Work-family conflict	8–57	Very Low < 15	123	19.46
		Low 15–25	210	33.23
		Medium 26–35	218	34.49
		High 36–45	64	10.13
		Very high > 45	17	2.69
Job burnout	15–105	Very low < 30	103	16.30
		Low 31–50	268	42.40
		Medium 51–70	236	37.34
		High 71–90	24	3.80
		Very high > 90	1	0.16
Affective commitment	3–21	Very Low < 10	6	0.95
		Low 10–15	170	26.90
		Medium 16–20	299	47.31
		High > 20	157	24.84
Safety participation	3–21	Very low < 14	61	9.65
		Low 14–17	161	25.47
		Medium 18–20	217	34.34
		High > 20	193	30.54
Safety compliance	3–21	Very low < 14	1	0.16
		Low 14–17	72	11.39
		Medium 18–20	246	38.92
		High > 20	313	49.53

**Table 4 ijerph-19-11056-t004:** The results from the mediation analyses.

Independent Variables	Dependent Variables								
	Job Burnout	Affective Commitment		Safety Participation			Safety Compliance		
	M1	M2-1	M2-2	M3-1	M3-2	M3-3	M4-1	M4-2	M4-3
1. Self-esteem	−0.44 ***	0.24 ***	0.01	0.37 ***	0.15 ***	0.15 ***	0.28 ***	0.14 ***	0.14 ***
2. Work-family conflict	0.41 ***	−0.28 ***	−0.06	−0.17 ***	0.03	0.06	−0.15 ***	−0.02	0.01
3. Job burnout			−0.54 ***		−0.50 ***	−0.25 ***		−0.31 ***	−0.17 ***
4. Affective commitment						0.47 ***			0.26 ***
Adjusted R^2^	0.48 ***	0.19 ***	0.34 ***	0.21 ***	0.34 ***	0.49 ***	0.13 ***	0.18 ***	0.22 ***
∆R^2^	0.48 ***	0.19 ***	0.15 ***	0.21 ***	0.13 ***	0.15 ***	0.13 ***	0.05 ***	0.05 ***

Notes: *** *p* < 0.001.

**Table 5 ijerph-19-11056-t005:** Results of the chain mediating effect based on the bootstrapping test.

Indirect Effects	Effect	Boot SE	95%CI	
			Boot LLCI	Boot ULCI
Dependent variable: Safety participation				
Work-family conflict --> Job burnout --> Safety participation	−0.10 ***	0.02	−0.14	−0.06
Work-family conflict --> Affective commitment --> Safety participation	−0.02	0.02	−0.06	0.01
Work-family conflict --> Job burnout --> Affective commitment --> Safety participation	−0.10 ***	0.02	−0.14	−0.08
Dependent variable: Safety compliance				
Work-family conflict --> Job burnout --> Safety compliance	−0.07 ***	0.02	−0.11	−0.03
Work-family conflict --> Affective commitment --> Safety compliance	−0.02	0.01	−0.04	0.01
Work-family conflict --> Job burnout --> Affective commitment --> Safety compliance	−0.06 ***	0.01	−0.09	−0.04

Notes: *** *p* < 0.001.

## Data Availability

Restrictions apply to the availability of these data. Data were obtained from Chengdu Rail Transit Group Co., Ltd. and are available from Zizheng Guo with the permission of Chengdu Rail Transit Group Co., Ltd.

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
