# Peer review of "How Work-Family Conflict Influenced the Safety Performance of Subway Employees during the Initial COVID-19 Pandemic: Testing a Chained Mediation Model"

_ijerph, 2022, doi:10.3390/ijerph191711056_

Round 1

Reviewer 1 Report

Dear Editor,

Thank you for your invitation to evaluate this manuscript. I have read and completed my review regarding the manuscript entitled "How work-family conflict can influence the safety performance of subway employees during the initial COVID-19 pandemic: Testing a chained mediation model".

The abstract section is fine, the number of the participants are quite satisfactory ( as it was stated in the manuscript "Using questionnaire data from 632 Chinese subway employees during February 2020, 22 structural equation modeling analyses were performed" ). Athough the data is relatively old (the data were collected in 2020 and we are in the second half of the 2022), it is not a big problem. The results are still valuable. The main problem regarding the manuscript is methodology. The author stated in the absract that they performed structural equation modeling analyses. And they examined the relationships between work-family conflict, job burnout, safety performance and affective commitment.The problems about the methodology can be listed as following:

1) There is no info or analyses regarding the normal distribution of the data. Please provide some proof if the data is normally distritbuted or not.

2) I could not see the confirmatory factor analyses (CFA) results. Please perform confirmatory factor analyses with your own data and show the results in the manuscript either in tables or in seperate paragraphs.

3) I recommend the authors to add AVE analyses and discriminant analyses results.

4) Please add a sepaerate heading for the limitations.

5) The implications of the study seems very weak. This paper deserves to have a detailed implications both for some practical tips for policy makers and further researches.

Reviewer 2 Report

Dear authors, thank you for the opportunity to get acquainted with your work. Your research is interesting, relevant, thoughtful and justified.

The introduction describes the research problem in detail and substantiates the author's approach, the purpose and hypotheses of the research.

Materials and methods describe the research procedure, research methods are described in detail and meet the purpose of the research. At the same time, it is necessary to expand the description of the sample in terms of the distribution of employees by groups with different length of service (not only indicate the average length of service), indicate the distribution by position or professional group, by level of education, by marital status (which is important when analyzing the existing problem). It is also possible to describe the mode of work and rest in which employees work and data on production factors, in addition to the pandemic. These data will allow better interpretation of the results of the study.

The results of the study are presented in detail and qualitatively illustrated. The conclusions are justified. At the same time, it is necessary to supplement descriptive statistics with an indication of the distribution of employees in % according to the level of burnout, conflict between work and family, commitment and safety (for a direct understanding of the severity of these factors among metro workers).

The discussion of the results is presented by a good overview of the correlation of the results of this study with the results obtained by other authors. Reveals the possible mechanisms of the obtained relationships. At the same time, the limitations of the study should be more clearly indicated (it is possible to single out a subsection), as well as practical recommendations based on the results of the study (a subsection can also be made).

In the conclusion, include more conclusions regarding the actual results of the study (model validation).

In general, the recommendations presented do not reduce the overall positive impression of the work.

Best regards and wishes for further success, the reviewer.

Round 2

Reviewer 1 Report

I examined the revised version of the manuscript entitled "How work-family conflict can influence the safety performance of subway employees during the initial COVID-19 pandemic: Testing a chained mediation model". I am glad to see that the authors accepted my recommendations and they revised the manuscript.  In response letter I saw the normality check results as Q-Q plot figure but I could not see it in the main manuscript text. Please add the figure showing the normality results in the main text. 

I have seen that the authors made considerable changes and revision on the manuscript. My personal opinion is to accept the manuscript after the minor revision. I do not need to see the manuscript again.